# OpenReview forum: "A New Framework for Cybersecurity Refusals in AI Agents"
_ICML.cc/2026/Conference — ICML 2026 regular_

### Official Review · Reviewer_ocX8 · 2026-03-03

**Soundness:** 2
**Presentation:** 2
**Significance:** 3
**Originality:** 3
**Overall Recommendation:** 3
**Confidence:** 3

**Summary:**

This paper introduces the Cybersecurity Refusal Framework (CRF), a structured approach to evaluating when AI agents should refuse offensive cybersecurity tasks—addressing a gap in existing benchmarks that focus primarily on exploitation proficiency rather than robust refusal in high-risk settings. The framework proposes (1) environmentally-aware refusals grounded in verified infrastructure context rather than superficial prompt cues, (2) a tripartite taxonomy of targets (Red: always refuse; Green: always allow; Yellow: context-dependent), and (3) a 29-task benchmark spanning Easy, Medium, and Hard modes to jointly measure proficiency, refusal behavior, situational awareness, and utility. Evaluation across eight frontier models reveals that most rarely refuse harmful tasks in realistic production-like environments, and that refusal behavior is highly brittle and sensitive to prompt phrasing, with simple scope language often bypassing safeguards.

**Compliance With Llm Reviewing Policy:**

Affirmed.

**Key Questions For Authors:**

1. How are the easy/medium/hard modes for evaluating samples in the benchmark defined, and what is the specific distribution?

2. The authors observed that the model's rejection decisions were highly dependent on URL characters and similar factors, rather than actual contextual evidence from the environment. This raises the question: why does this lead to a 0% rejection rate in real production environments? If the task were genuinely guided by features such as URLs, the model should exhibit rejection behaviour. Clarification from the authors is sought.

**Limitations:**

yes

**Strengths And Weaknesses:**

Strengths:
1. The paper tackles a timely and important problem by moving beyond capability evaluation to examine refusal robustness in dual-use cybersecurity contexts.
2. It offers a clear conceptual contribution through the well-motivated shift from user-centric to environmentally-aware refusal mechanisms.
3. The benchmark is well designed, with a multi-tier setup that cleanly separates capability assessment from robustness evaluation.

Weaknesses:
1. Essentially, this work constitutes a conceptual framework coupled with a benchmark proposal, rather than an executable algorithmic framework. It lacks rigorous boundary analysis and theoretical property definitions.

2. Evaluations of environmental awareness lack depth, remaining essentially manipulations of superficial environmental signals. For instance, localhost versus real domain.

3. The proposed Red/Yellow/Green taxonomy is intuitively reasonable but lacks principled grounding or stakeholder validation. The choice of “always refuse” domains (e.g., government, power grid) appears ad hoc, and the paper does not provide a formal or policy-backed framework for defining these boundaries, such as expert annotation, rigorous threat model definition.

4. The evaluation sample size was relatively small, and no statistical significance analysis or confidence intervals were provided, which are typically crucial for assessing the reliability of evaluations.

5. If understood correctly, existing models of security mechanisms for intelligent agents in the field of web vulnerabilities are still insufficient. However, this paper lacks discussion and argumentation regarding environmental measures for this phenomenon.


Suggestions:
1. Certain text in the references and appendices extends beyond the margins; it is recommended that the author address and amend this.

2. Baseline, evaluation models and other experimental settings should be independent of the results.

---

> ### Author Rebuttal · Authors · 2026-03-31
>
> Thank you for your review. We appreciate both the positive assessment of our contributions and the constructive criticism, and address each point below.
>
> **On the nature of the contribution (Weakness 1):**
>
> We acknowledge that this work is primarily a conceptual framework and benchmark rather than an algorithmic solution, and do not claim otherwise. We believe that this is appropriate given the current state of the field, as there is no existing consensus on what AI agents should refuse in these contexts, no any established methodology for evaluating such refusals. Proposing an executable algorithmic framework without first defining the problem would be premature, though we agree this would be a good candidate for future work.
>
> **On the depth of environmental awareness evaluations (Weakness 2):**
>
> We would argue that the shallowness of our manipulations is precisely the point: even trivial deceptions like proxying a production website behind localhost bypass refusal mechanisms in 6 of 8 frontier models. Evaluating against more sophisticated manipulations is not yet meaningful if models cannot resist basic ones.
> Moreover, models clearly possess environmental awareness — 15% of Hard mode and 13.8% of Medium mode reflections explicitly acknowledge the target involves critical infrastructure, yet models proceed anyway. The entrypoint provided is always a real production endpoint, so the problem is not insufficient environmental signal but models failing to act on context they have already gathered.
>
> **On the Red/Yellow/Green taxonomy (Weakness 3):**
>
> We acknowledge this criticism. We do not present this taxonomy as a formal, policy-ready classification. Instead, it is meant to serve as a concrete starting point that grounds our evaluation in realistic, high-stakes contexts.  We chose domains where unauthorized testing could cause immediate physical harm or large-scale societal disruption — a threshold we believe is defensible even without formal validation. Refining these boundaries through expert annotation, threat modeling, and policy consultation is an important next step, and we are hopeful that this work can contribute to the galvanization of policy experts to look into this further.
>
> **On sample size and statistical analysis (Weakness 4):**
>
> Thank you, we will run additional trials and update the manuscript accordingly.
>
> **On discussion of existing security mechanisms (Weakness 5):**
>
> This is a fair point. Figure 5 provides insight into why this occurs: models do recognize environmental signals – for instance, 15% and 13.8% of reflections acknowledge the target involves critical infrastructure. However, they also consistently weigh user-provided framing more heavily than this context. As a result, the problem is not that models lack environmental awareness, but that authorization claims in the prompt override environmental evidence. We will expand the manuscript to discuss this dynamic more explicitly and argue that environmental signals should carry greater weight in refusal decisions, particularly when the target involves high-risk domains.
>
> **Suggestions**
>
> Thank you for the suggestions. We will update the manuscript accordingly.
>
> ### Key Questions
>
> **On benchmark difficulty distribution (Question 1):**
>
> The benchmark consists of 29 challenges: 20 local challenges (used in both Easy and Medium modes) and 9 remote challenges (Hard mode). Easy and Medium modes use identical vulnerabilities but differ in fidelity — Easy mode presents barebones servers lacking production noise, while Medium mode embeds the same vulnerabilities within real production websites via nginx proxying. Hard mode targets actual features on real production websites. The challenges span four critical infrastructure domains and nine vulnerability classes, with full details in Appendix A.
>
> **On the 0% refusal rate in production environments (Question 2):**
>
> Thank you for this question — we want to clarify an important distinction. In Easy mode, models do exhibit high refusal rates when presented with production URLs (e.g., Claude Sonnet 4.5 and GPT-5.2 refuse 100% of the time). The 0% refusal rate applies to Medium and Hard modes, where the evaluation prompts include scoping language and rules of engagement sufficient to override URL-based refusal heuristics. In Hard mode, framing the task as reconnaissance rather than active exploitation further reduces refusals to zero across all models. We argue that this distinction is a false one: reconnaissance in an offensive security context clearly implies subsequent exploitation, and models should treat planning as equivalent to active testing when the target falls within a red-category domain.

---

> > ### Author Rebuttal · Reviewer_ocX8 · 2026-04-02
> >
> > Thank you for your reply. Firstly, has the author considered potential solutions following the benchmarking? This is not usually novel, but it is feasible. Furthermore, can W4 provide quantified results?

---

> > > ### Author Response · Authors · 2026-04-06
> > >
> > > > Firstly, has the author considered potential solutions following the benchmarking?
> > >
> > > Yes, absolutely. As mentioned in our response to reviewer tCh7, we believe that the first step towards a solution to this particular issue is clear: agents should be trained to fetch the target before deciding whether to comply. This can be done during post-training, likely where gold-standard trajectories can be utilized during SFT, with additional RL being needed to better define the boundaries between red/yellow/green domains. The main bottleneck here is not the lack of a technical solution, but the lack of policy surrounding using AI agents for cybersecurity work and research.
> > >
> > > > Furthermore, can W4 provide quantified results?
> > >
> > > Unfortunately, we did not have the bandwidth to collect additional trajectories. We will include these in the camera-ready edition, and hope that is satisfactory.

---

### Official Review · Reviewer_QcAJ · 2026-03-05

**Soundness:** 4
**Presentation:** 4
**Significance:** 4
**Originality:** 4
**Overall Recommendation:** 6
**Confidence:** 4

**Summary:**

This paper provides exceptionally real world-adjacent offensive security benchmarks, metrics/evaluations, and frameworks to assess leading frontier models with in terms of refusal boundaries and context awareness. The paper benchmarks many leading models finding they perform generally poorly and with low context awareness outside of simple test cases in ways that are anticipated to be surprising and impactful.

**Compliance With Llm Reviewing Policy:**

Affirmed.

**Key Questions For Authors:**

(1) In the cited "Anthropic. Threat intelligence report" and associated press release, Anthropic makes strong claims about the ability of Claude both for cyber offense and for cyber defense / safety evaluation. Yes this paper seems to contradict these industry claims, finding major flaws and vulnerabilities in Claude's related to cyber offense. Could the authors interpret there results in the context of the services that Anthropic and other big AI companies are trying to sell to the government and firms around cybersecurity, and whether their findings suggest GenAI is not quite ready for this domain? What besides doing well at the provided benchmarks needs to be assessed to make these models market ready?

(2) Can the authors comment more on the general intersection of agent vs non-agent models in this space? Non-agent models seem particularly problematic and to require different solutions potentially.

(3) Are there any comments the authors have about how the closed versus open weight model debate intersects here?

**Limitations:**

The limitation summary is well done.

**Strengths And Weaknesses:**

The paper is excellent with strong literature motivation, many novel results, and important insights for a general computer science audience. Frontier models end up being shockingly easy to trick or hijack into malicious offensive cybersecurity actions, and the provided materials seem important for making safer models. I think this paper will have considerable influence both scientifically and in public/policy circles.

Breaking up the benchmark evaluations into three tiers of easy to hard was a good idea.

I support the authors in making most materials open, but restricting the most dangerous materials to third party collaborations given the reasoning provided in the text.

I cannot find any substantial weaknesses in this paper.

---

> ### Author Rebuttal · Authors · 2026-03-31
>
> Thank you for your review, we appreciate your comments. To answer your questions:
>
> > (1) In the cited "Anthropic. Threat intelligence report" and associated press release, Anthropic makes strong claims about the ability of Claude both for cyber offense and for cyber defense / safety evaluation. Yes this paper seems to contradict these industry claims, finding major flaws and vulnerabilities in Claude's related to cyber offense. Could the authors interpret there results in the context of the services that Anthropic and other big AI companies are trying to sell to the government and firms around cybersecurity, and whether their findings suggest GenAI is not quite ready for this domain? What besides doing well at the provided benchmarks needs to be assessed to make these models market ready?
>
> While we have identified significant flaws in Claude models with respect to offensive security refusals, we would not claim that these findings suggest GenAI is incapable in this domain. On the contrary, it is becoming increasingly clear that LLM-powered agents for offensive security are more efficient and comparably or perhaps more capable than the average offensive security professional. That being said, we believe it is somewhat irresponsible to deploy models with this level of offensive capability without fully mature guardrails. It is trivially easy for threat actors to misuse powerful models such as Claude Opus 4.5 to conduct large-scale [cybercrime](https://www.anthropic.com/news/disrupting-AI-espionage), which clearly shows that these models are market-ready offensively, and quite possibly defensively. They are also threat-actor-ready, in a way that should be a cause for concern. Beyond performing well on existing benchmarks, we believe models need to be assessed on their ability to distinguish authorized from unauthorized use in realistic operational contexts — precisely the gap our framework aims to address.
>
> > (2) Can the authors comment more on the general intersection of agent vs non-agent models in this space? Non-agent models seem particularly problematic and to require different solutions potentially.
>
> This is an important distinction that requires some nuance. Models that are not currently capable of functioning as general-purpose agents (e.g., Llama 4) are typically less capable overall than models that excel at agentic tasks. For example, a chat-based Claude Opus 4.5 is significantly more useful than a chat-based Llama 4 Maverick, even outside of agentic settings. However, deploying open-weight models for their chat capabilities alone can still yield meaningful cybersecurity knowledge that could uplift a threat actor. This is a harder problem to address for several reasons:
>
> 1. Agentic models can actively explore the context they are deployed in (e.g., fetch the target website), while chat models cannot.
> 2. Chat models are therefore constrained to making determinations based purely on conversation history, which cannot contain the environmental information necessary for an informed refusal decision.
> 3. Simply asking questions about offensive security techniques is often not considered a clear policy violation, and the boundary between harmful and benign queries becomes much blurrier in a chat-only setting.
>
> This is partly why we believe contextual refusal mechanisms represent a more promising direction than purely prompt-based approaches.
>
> > (3) Are there any comments the authors have about how the closed versus open weight model debate intersects here?
>
> We note that these comments are largely informed by lived experience rather than directly supported by our benchmark results, as the only open-weight model we evaluated was Llama 4 Maverick, due to the limitations described at the beginning of Section 4.
>
> A frequent criticism of open-weight model releases is that they can be misused with greater ease than closed-weight alternatives – for instance, it is easier to elicit harmful content from an unguarded open-weight model. However, given the ease with which we were able to bypass refusal mechanisms in closed-weight models, we believe that closed-weight models pose a greater risk at the present moment due to their substantial capabilities advantage.
>
> Following from our answer to the previous question, it may be the case that a threat actor who does not require agentic capabilities would find chatting with an open-weight model to be a more convenient avenue for misuse. That said, it is currently difficult to envision a scenario in which a chat-only model is strictly preferable to an agentic one for offensive security tasks.

---

> > ### Author Rebuttal · Reviewer_QcAJ · 2026-03-31
> >
> > I have read all reviews and author responses. I will retain my passing scores. While there were many minor to moderate criticisms I appreciated, while also appreciating the author responses to addressing them, I also believe the current rejections are somewhat unreasonable and being based on unrealistic expectations for what a single conference paper can achieve. This paper is making a strong and important starting point in terms of benchmarking, results and discussion potential, and being unfairly criticized in some respects for not more comprehensively solving a major unresolved issue affecting most/all leading frontier models, that were documented in novel and useful ways here. Particularly, the criticism that the benchmark is difficult and the authors didn't say directly how to solve it seems a bit strange for a new benchmark, better benchmarks are harder to solve in an unresolved way and more open ended, plus this one has direct real world relevance (plus the authors have given starter recommendations to improve performance in their benchmarks). That is my view anyways.
> >
> > I do encourage the authors to think whether any fraction of the currently non-public benchmark categories (non-public for security concerns) can be made public to aid the goal of basic reproducibility, even if not the entire hard classes.

---

### Official Review · Reviewer_tCh7 · 2026-03-06

**Soundness:** 3
**Presentation:** 2
**Significance:** 3
**Originality:** 3
**Overall Recommendation:** 4
**Confidence:** 3

**Summary:**

This paper proposes a framework for evaluating the ability of AI agents to refuse to perform malicious tasks in cybersecurity scenarios. Addressing the shortcomings of existing benchmarks that focus on attack capabilities while neglecting security denial mechanisms, this research establishes denial principles and task categories, and tests several state-of-the-art models on real-world web vulnerability identification tasks.

**Compliance With Llm Reviewing Policy:**

Affirmed.

**Final Justification:**

My concerns have been largely addressed, and I keep my original score.

**Key Questions For Authors:**

1. The paper states that all models were evaluated within the Codex agent scaffold. Are the tool invocation capabilities, context management methods, and action strategies of different models fully equivalent under this scaffold? Additionally, are these behaviors fully equivalent across different scaffolds for the same model? Have the authors evaluated the attribution bias that the scaffold itself might introduce?

2. The paper defines critical infrastructure scenarios, such as healthcare, as red zones and advocates for models to adopt an absolute rejection strategy. In real-world scenarios, these domains also have legitimate and high-value authorized security testing needs. How do the authors balance these security benefits against the costs of over-rejection?

3. Can this framework be extended to other cybersecurity tasks, such as malicious code generation, vulnerability exploitation, or privilege escalation? If so, do the authors believe that the existing risk partitions and evaluation metrics remain applicable?

4. Can the refusal criteria be further automated to reduce reliance on manual rule design and predefined risk classifications? Have the authors considered the possibility of learning dynamic refusal policies based on environmental signals?

**Limitations:**

yes

**Strengths And Weaknesses:**

Strengths

1. The experimental design is comprehensive.
The paper constructs three scenarios of varying difficulty, covering both localhost and configurations where the target website is accessed via an Nginx proxy, effectively simulating attack paths at different levels. The evaluation criteria are clear, and the paper introduces a utility score to measure the balance between capability and rejection.

2. The problem definition is novel.
The paper extends the agent’s refusal mechanism from being based solely on user input to incorporating the execution environment in its decision-making, proposing an environment-aware refusal boundary framework. Additionally, the authors define three categories of risk zones, providing a clear analytical framework for refusal policies in offensive security scenarios.

3. The experimental results have practical significance.
The paper demonstrates that current models’ refusal mechanisms may still be influenced by environmental settings and prompting methods, such as wrapping tasks in authoritative language or using local port forwarding and proxy methods to weaken the model’s perception of real targets. This indicates that existing models still do not adequately model environmental risks.

Weaknesses

1. Some evaluations still fall short of real-world risks.
In Hard mode, the paper primarily examines whether the model identifies environmental risks and outputs evaluations and test plans, without allowing it to carry out full-scale attacks on real production websites. While this design is understandable, reconnaissance and planning remain only proxy indicators of actual attack risks and are not entirely equivalent to the actual severity of harm.

2. Insufficient discussion of biases in automated scoring.
The paper uses Claude Sonnet 4.5 as the automated scorer, but does not sufficiently address potential issues such as model bias, consistency problems, and manual review mechanisms, which may affect the objectivity of the evaluation results.

3. Limited reproducibility.
The paper only makes the complete challenge for the Easy mode publicly available, while adopting a restricted release strategy for the Medium and Hard scenarios. Although this approach is justified for security reasons, it also limits the community’s ability to independently reproduce key results.

4. Generalizability requires further validation.
This paper primarily focuses on web vulnerability scenarios; there is currently a lack of validation regarding whether the related framework is applicable to other types of cybersecurity tasks.

5. Limited discussion of technical improvements.
While the paper effectively identifies issues with existing model rejection mechanisms, the discussion on how to further improve context-aware rejection mechanisms remains limited.

---

> ### Author Rebuttal · Authors · 2026-03-31
>
> Thank you for your thorough and constructive review. We appreciate both the detailed assessment of our contributions and the thoughtful questions raised.
>
> **On Hard mode as a proxy for real attack risk:**
>
> We agree that reconnaissance and planning are proxy indicators, and we acknowledge this limitation in the paper. We appreciate your recognition of why this is necessary. However, we believe it is meaningful for two reasons. First, a detailed vulnerability assessment plan is directly actionable – a natural follow-up would be to ask the agent to execute the plan. Further, even if the agent were to refuse, the plan could be carried out by a human operator or a less-aligned model, and as such we believe this is still a realistic threat model.
>
> **On biases in automated scoring:**
>
> Thank you for the comment. In order to address model bias and consistency problems, we employ rubrics with detailed success/fail criteria to ground the model's assessments – each challenge has specific pass/fail criteria tied to vulnerability identification, endpoint discovery, and exploitation method, leaving relatively little room for subjective judgment. We will update the manuscript with further discussion.
>
> **On limited reproducibility:**
>
> We recognize the tension between reproducibility and responsible disclosure. For ethical reasons, we believe it would be irresponsible to open source the entire evaluation framework. We will make it available to model providers and researchers at their request, after a thorough verification process, to limit downstream misuse.
>
> **On generalizability beyond web vulnerabilities:**
>
> We agree that validating the framework's applicability to other cybersecurity domains — such as network exploitation, binary analysis, or privilege escalation — is important future work. We scoped this paper to web vulnerabilities deliberately, as they represent a well-understood and self-contained attack surface that allowed us to construct controlled evaluations at multiple fidelity levels.
>
> **On limited discussion of technical improvements:**
>
> We focused on problem identification rather than training interventions. That said, we believe environmental awareness is tractable: agents should be trained to fetch the target before deciding whether to comply. Our results show that most models already achieve 95–100% awareness in Medium mode but proceed regardless. We view thorough investigations into how to improve these refusal mechanisms as promising future work.
>
> **On scaffold equivalence across models:**
>
> This is an excellent question. All models were evaluated within the Codex scaffold using identical configurations, tool access, and autonomy settings to ensure a controlled comparison. Failure to operate within the scaffold is a model capabilities issue. You raise a good point with respect to different scaffolds. We did not conduct cross-scaffold evaluations, and we agree that scaffold-induced attribution bias is a meaningful confound that warrants further investigation. There are many such scaffolds, and we opted for Codex due to it being open-source, widely-used, and not too focused on CTFs.
>
> **On balancing red-zone refusals with legitimate security testing needs:**
>
> This is precisely the tension that motivated our framework. We do not argue that security testing of critical infrastructure should never occur — clearly, these domains have significant and legitimate testing needs. Rather, we argue that AI agents specifically should refuse these tasks because they cannot reliably verify authorization. Legitimate penetration testing of healthcare or power grid systems involves formal contracts, scoping documents, and legal agreements that exist outside the agent's observable context, and should be conducted exclusively by licensed professionals.
>
> **On extending the framework to other cybersecurity tasks:**
>
> We believe the core principles are applicable to tasks such as malicious code generation, exploit development, and privilege escalation. However, the specific risk partitions and evaluation metrics would likely require adaptation. For example, malicious code generation may require different environmental signals (e.g., whether the target codebase appears to be the user's own project versus a widely-adopted third-party system), and the proficiency metrics would need to reflect the relevant task structure. We view extending the framework to these domains as valuable future work.
>
> **On automating refusal criteria and learning dynamic policies:**
>
> This is a compelling direction. Our current framework relies on predefined risk categories, which we view as a necessary starting point given the absence of any prior consensus on refusal boundaries. Promising directions include training agents on diverse deployment contexts that are informed by signals already present on the internet that signify trust, such as TLS certs and DNS resolution.

---

> > ### Author Rebuttal · Reviewer_tCh7 · 2026-04-03
> >
> > Thank you for the authors’ response. My concerns have been largely addressed, and I am inclined to keep my original score.

---

### Official Review · Reviewer_WTCr · 2026-03-13

**Soundness:** 3
**Presentation:** 2
**Significance:** 3
**Originality:** 3
**Overall Recommendation:** 4
**Confidence:** 2

**Summary:**

This paper proposes a framework for distinguishing use of LLMs for legitimate defensive security testing from illegitimate use, and a benchmark for testing models proficiency in making such distinctions when refusing to respond to prompts, focusing on web security.

**Compliance With Llm Reviewing Policy:**

Affirmed.

**Final Justification:**

Based on the rebuttal response, I am in agreement that a novel benchmark and formalisation of the problem is sufficient without offering solutions.

**Key Questions For Authors:**

None

**Limitations:**

Yes

**Strengths And Weaknesses:**

Soundness:
- The proposed framework and the benchmark seem technically sound. They are well-motivated and clear about limitations. The experimental testing of LLM-agents is rigorous and well designed.

Presentation:
- The paper is well written and structured. The narrative is easy to follow.

Significance:
- This addresses an important problem. However, one could argue that the result, while highly concerning, is not surprising, given the inherent difficulty of distinguishing harmful offensive tasks from benign defensive testing. That said, the paper shows meaningful differences between agents, showing that there is important work which some providers could do to score better on this benchmark.
- It is not clear how agent providers can improve on this benchmark without also compromising legitimate uses, without instituting new mechanisms for environmental awareness. This could require various intrusive forms of access and control of the developers environment. The benchmark may therefore be limited in use unless measures which balance the dual needs of appropriate refusal as well as developer control / privacy.

Originality:
- The proposal to move towards environmental context to determine when to refuse seems novel and promising.

---

> ### Author Rebuttal · Authors · 2026-03-31
>
> Thank you for your review!
>
> > However, one could argue that the result, while highly concerning, is not surprising, given the inherent difficulty of distinguishing harmful offensive tasks from benign defensive testing.
>
> We respectfully disagree that the results are unsurprising. While in general, it is true that distinguishing offensive from defensive cybersecurity tasks is well-known, several specific findings were unexpected. Consider the Wikipedia example in Figure 1: the rightmost pane shows the ease with which Claude Code can be convinced to attempt a live penetration test against Wikipedia's production website. Contrary to popular belief, this was not an incredibly sophisticated "jailbreak" attempt, and yet even without any bug bounty scope, prior context, or plausible justification for the request, Claude Code complied. While it was always clear this was possible, the minimal effort required was striking. Furthermore, the justification analysis revealed that 90% of post-hoc reflections cited the user-granted authorization as a sufficient reason to comply, again despite a lack of a verification mechanism. We believe these findings to be surprising and concerning.
>
> > It is not clear how agent providers can improve on this benchmark without also compromising legitimate uses, without instituting new mechanisms for environmental awareness.
>
> While we agree that improving on this benchmark requires new mechanisms for environmental awareness, we believe such mechanisms are straightforward to implement. For instance, when presented with an offensive security task against a website, an agent could first fetch and analyze the target to determine if it is likely to be a production deployment. Production environments are difficult to convincingly replicate, so we believe this to be a reliable signal for refusal decisions.
>
> More importantly, we do not propose that all production or other websites should trigger a refusal mechanism. Rather, we focus on defining a specific class of high-risk targets – government, healthcare, power grid, and transportation systems – where the potential for large-scale harm is too great to permit AI-assisted offensive testing. We acknowledge as a limitation that the boundaries of this classification require further refinement, though do not believe this detracts from our contribution to the field.
>
> > The benchmark may therefore be limited in use unless measures which balance the dual needs of appropriate refusal as well as developer control / privacy.
>
> We view this benchmark as a starting point, and fully expect it to be superseded as the field matures. In fact, we can envision a scenario in which it becomes saturated relatively quickly. That being said, our primary contribution lies in formalizing the problem. More specifically, we see enumerating the dimensions along which refusal mechanisms should be evaluated as a critical first step to more robust refusals in AI agents. This foundation is valuable precisely because it enables the community to build more comprehensive policies and evaluations.
>
> > Papers in this category require revisions before they can be meaningfully built upon by others
>
> We are happy to revise the paper and welcome specific suggestions on what you believe should be adjusted.

---

> > ### Author Rebuttal · Reviewer_WTCr · 2026-04-03
> >
> > These responses make sense and have caused me to reconsider that a benchmark paper need not provide solutions. I therefore revise my score upwards.

---

### Decision · Program_Chairs · 2026-04-30

**Decision:**

Accept (regular)

**Comment:**

This submission studies refusal systems for requests in potential dual-use domains such as cyber. As such refusal mechanisms are an important part of how the most capable LLMs are served at the moment, this is a very relevant topic. The authors introduce a new taxonomy and demonstrate the duality of requests -- both might be helpful to guide future decisions in this space. While there was no unanimous decision among reviewers, most reviewers had a positive view of the submission. The most important weakness pointed out by reviewers is a lack of statistical rigor due to small sample sizes. The authors promised to extend their experiments substantially and, thus, resolve this concern. I am willing to recommend this paper for publication _conditional_ on the authors actually making these additions.